

# On the stability of the infinite Projected Entangled Pair Operator ansatz for driven-dissipative 2D lattices

**Dainius Kilda[1]⋆, Alberto Biella[2,3], Marco Schiró[3,4],
Rosario Fazio[5,6] and Jonathan Keeling[7]**

**1** Division of Chemistry and Chemical Engineering,
California Institute of Technology, Pasadena, CA, USA
**2** Université Paris-Saclay, CNRS, LPTMS, 91405 Orsay, France
**3** JEIP, USR 3573 CNRS, Collège de France, PSL Research University,
11 Place Marcelin Berthelot, 75321 Paris Cedex 05, France
**4** Institut de Physique Théorique, Université Paris Saclay,
CNRS, CEA, 91191 Gif-sur-Yvette, France
**5** Abdus Salam ICTP, Strada Costiera 11, I-34151 Trieste, Italy
**6** Dipartimento di Fisica, Università di Napoli 'Federico II',
Monte S. Angelo, I-80126 Napoli, Italy
**7** SUPA, School of Physics and Astronomy, University of St Andrews,
St Andrews, KY16 9SS, United Kingdom

⋆ dainius.kilda@gmail.com

## Abstract

We present calculations of the time-evolution of the driven-dissipative XYZ model using the infinite Projected Entangled Pair Operator (iPEPO) method, introduced by [A. Kshetrimayum, H. Weimer and R. Orús, Nat. Commun. 8, 1291 (2017)]. We explore the conditions under which this approach reaches a steady state. In particular, we study the conditions where apparently converged calculations may become unstable with increasing bond dimension of the tensor-network ansatz. We discuss how more reliable results could be obtained.

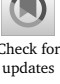

# 1 Introduction

Tensor network approaches have provided a route to efficient numerical simulations across a wide range of physical problems [1–4]. In one dimension, matrix product states (MPS) were originally introduced in the context of one-dimensional quantum ground states [5, 6]. They have subsequently been extended to finite temperatures [7, 8], and to open quantum systems and density-matrix evolution [9, 10]. Such methods have been very fruitful in exploring the nonequilibrium steady states (NESS) of driven-dissipative one-dimensional systems, using matrix product operators (MPO) [11–21]. These methods can also be extended beyond one dimension, either by mapping a finite two-dimensional lattice onto a one-dimensional chain [22]—see Ref. [21] for a driven-dissipative implementation—or via the projected entangled pair state (PEPS) algorithm [23–26]. The PEPS approach represents the two-dimensional lattice directly as a tensor network [3, 4], and allows a direct simulation of an infinite (translationally invariant) lattice (iPEPS).

In a significant development, Kshetrimayum et al. [27] presented results adapting the iPEPS algorithm to simulate open quantum systems on infinite 2D lattices. By using an infinite projected entangled pair operator (iPEPO) algorithm, they calculated the NESS of the dissipative XYZ and transverse field Ising models — i.e. finding the steady state of a many-body Lindblad master equation. The ability to routinely apply such methods to two-dimensional open quantum systems is potentially very powerful. While one-dimensional systems have shown a rich variety of collective behaviour, symmetry-breaking phase transitions generally do not occur in open one-dimensional systems, while they can in two dimensions. Several alternative approaches to approximately simulate two-dimensional open systems have been proposed, including cluster mean field theory [18], corner space renormalization [28, 29], and neural network states [30–34]. However, so far, these methods have generally been restricted to small systems (or small clusters), making it challenging to extract critical behavior. As such, the ability to routinely use iPEPO could be extremely powerful to numerically explore phase transitions and critical behavior in driven-dissipative systems.

Here, we explore in detail the stability of the iPEPO algorithm, introduced by Kshetrimayum et al [27]. We find that while at short times the algorithm shows reasonable time evolution, the behaviour at long times varies. In particular, we find that the algorithm only reaches a steady state in some parameter regimes, and close to dissipative critical points [35] it can fail to reach a steady state. The regimes where we fail to find a steady state correspond closely to the regimes where Ref. [27] found a larger value of their parameter $\Delta$, which measures how close the state they find is to a steady state. Moreover, we find that for some parameters, increasing bond dimension of the iPEPO representation does not systematically improve the accuracy of the results. On the contrary, it can in some cases destabilize a fixed point obtained at a lower bond dimension. We also suggest some possible alternatives to the simple-update iPEPO algorithm, which could help to alleviate the problem.

Our paper is organised as follows. In Sec. 2 we apply the iPEPO algorithm to calculate NESS

of the dissipative XYZ model in 2D, and analyse whether a steady state can be found. Section. 3 concludes with some comments on alternative tensor network approaches for computing NESS in 2D. We also provide an extended appendix, Sec. A, which summarises our implementation of the iPEPO algorithm; this implementation can be found at [36]. Since the core of the iPEPO and iPEPS algorithms is similar, we also present results benchmarking our implementation against the prototypical applications of iPEPS: the ground states of the transverse field Ising model and the hardcore Bose-Hubbard model.

## 2 Application to the dissipative XYZ model

In this section, using the iPEPO implementation described and benchmarked in Appendix A, we discuss finding the NESS of the dissipative spin-1/2 XYZ model on an infinite square lattice. The specific density-matrix equation of motion that we consider is:

$$\partial_t \rho = -i[H_{XYZ}, \rho] + \frac{\kappa}{2} \sum_j \left( 2\sigma_j^- \rho \sigma_j^+ - \sigma_j^+ \sigma_j^- \rho - \rho \, \sigma_j^+ \sigma_j^- \right), \tag{1}$$

$$H_{XYZ} = \sum_{\langle i,j \rangle} \left( J_x \sigma_i^x \sigma_j^x + J_y \sigma_i^y \sigma_j^y + J_z \sigma_i^z \sigma_j^z \right), \tag{2}$$

where $\sigma_j^{x,y,z}$ are Pauli matrices at lattice site $j$, $\sigma^{\pm} = \frac{1}{2}(\sigma^x \pm i\sigma^y)$, and $J_{x,y,z}$ are spin-spin coupling constants.

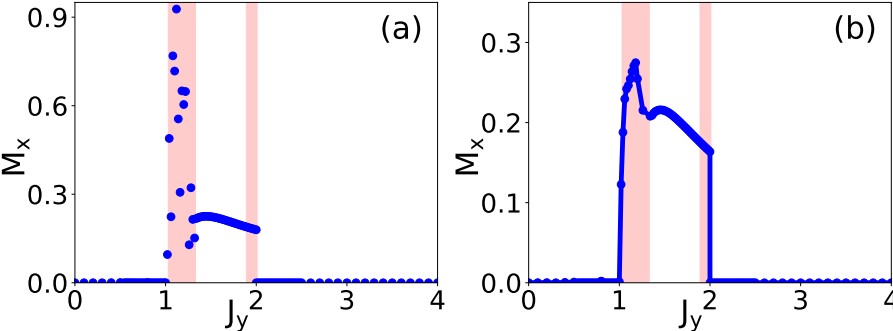

Figure 1: The magnetization order parameter $M_x$ of the dissipative XYZ model as a function of coupling strength $J_y$, for $J_x = 0.5$, $J_z = 1$ and $D = 4$. Energies given in units of $\kappa = 1$. The red highlighted areas indicate parameter regimes where the iPEPO algorithm fails to reach a steady state. (a) Results computed using timesteps $\delta t = 10^{-1}, 10^{-2}$ run until a steady state is found. (In the red regions, the run is stopped after $N = 1000$ steps with $\delta t = 10^{-1}$ followed by $N = 2000$ steps with $\delta t = 10^{-2}$.) (b) Results calculated using a large timestep $\delta t = 10^{-1}$ and stopping after $N = 1000$ steps.

We begin by computing the time evolution of the dissipative XYZ model for the same parameters considered by Ref. [27]. Figure 1(a) shows magnetization averaged over the two sites $i = A, B$ of iPEPO unit cell, $M_x = \frac{1}{2}(|\langle \sigma_{i=A}^x \rangle| + |\langle \sigma_{i=B}^x \rangle|)$, as a function of coupling strength $J_y$ using iPEPO bond dimension $D = 4$. We find that iPEPO algorithm only reaches a steady state for some values of $J_y$, while in the red highlighted areas no steady state is found—the results continue to change with time. Where a steady state is found, our results closely match Ref. [27]. The red regions in our figure—where no steady state is reached—correspond to points where the Kshetrimayum et al [27] report a large error in their steady state result. As observed in [27], these regions occur near the critical points, where one can expect correlation lengths to diverge. Figure 1(b) shows that if we use a large timestep, $\delta t = 10^{-1}$, and

deliberately stop the simulation early — i.e. after $N = 1000$ steps — one can reproduce results similar to those presented by Kshetrimayum et al [27] in the red region. However, because the simulation was stopped artificially early, the results in Fig 1(b) do not correspond to an actual steady state, and also inevitably contain significant Trotter errors due to the large timestep size. As we discuss further below, the failure to reach a steady state that we observe occurs specifically in the SU time evolution. That is, it is completely unaffected by the corner transfer matrix (CTM) contractions needed to compute observables. As a result, none of the results in the rest of this paper depend on the CTM contraction or environment bond dimension.

We have encountered similar issues of failing to reach a steady state in other parameter regimes of the dissipative XYZ model, as well as for other systems such as the dissipative transverse field Ising model in 2D. This raises important question about the practicality of the iPEPO algorithm as a tool to find the NESS of open quantum systems. In the following, to keep our discussion concise, we will restrict our attention to the dissipative XYZ model. Our goal below will be to understand when the iPEPO algorithm does and does not reach a steady state, focusing entirely on the SU time evolution, by studying the relative change of singular values. To measure this, we define

$$\epsilon_\Lambda = \frac{|\Lambda_n - \Lambda_{n-1}|_{\max}}{\delta t \, |\Lambda_n|_{\max}}, \tag{3}$$

in terms of the set of singular values $\Lambda_n$ at timestep $n$. As such, $\epsilon_\Lambda$ is a measure of the *largest* change of singular value, rescaled for ease of comparison between different timesteps.

## 2.1 Effects of simulation protocol

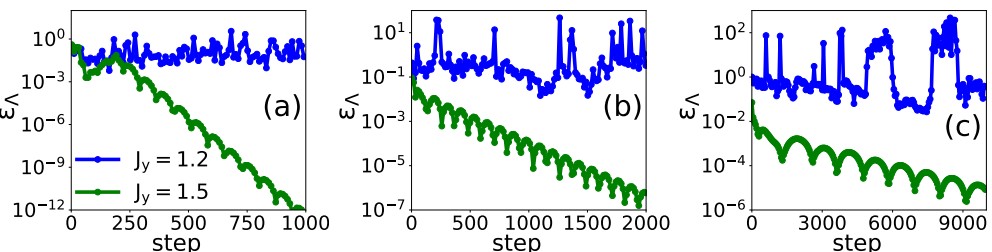

Figure 2: The evolution of $\epsilon_\Lambda$ (from $\Lambda^{[U]}$ as defined in Eq. (3)) at $J_y = 1.2$ (where no steady state is found) and $J_y = 1.5$ (where a steady state is found), using timesteps (a) $\delta t = 10^{-1}$, (b) $\delta t = 10^{-2}$, (c) $\delta t = 10^{-3}$. Energies given in units of $\kappa = 1$.

In Figs. 2(a-c) we plot $\epsilon_\Lambda$ against simulation step, for timestep sizes $\delta t = 10^{-1}, 10^{-2}, 10^{-3}$ respectively. We show both $J_y = 1.2$, in the parameter regime where iPEPO fails to reach a steady state (blue line), and $J_y = 1.5$, where a steady state is found (green line). At $J_y = 1.5$, we observe clearly that $\epsilon_\Lambda$ quickly decreases, indicating that we approach a steady state. However, at $J_y = 1.2$ we see $\epsilon_\Lambda$ undergoes noisy oscillations throughout the time evolution, for all timestep sizes, never approaching zero.

We next explore if using different initial conditions affects whether a steady state is found. Figures 3(a-c) show that the evolution of $\epsilon_\Lambda$ at $J_y = 1.2$ remains noisy for various initial conditions: a random number state (brown line), a state with all spins pointing 'down' (red line), a state with all spins pointing 'up' (orange line), and a state where each spin has components $\langle \sigma^x \rangle = 1, \langle \sigma^y \rangle = 1, \langle \sigma^z \rangle = -1$ (green line). Other initial conditions that we have tested (not shown here) produced a similar behaviour as in Fig. 3.

Another possible way to choose initial conditions for the problematic parameter regime is an adiabatic parameter sweep. We first calculate the NESS for a value of $J_y$ where iPEPO does

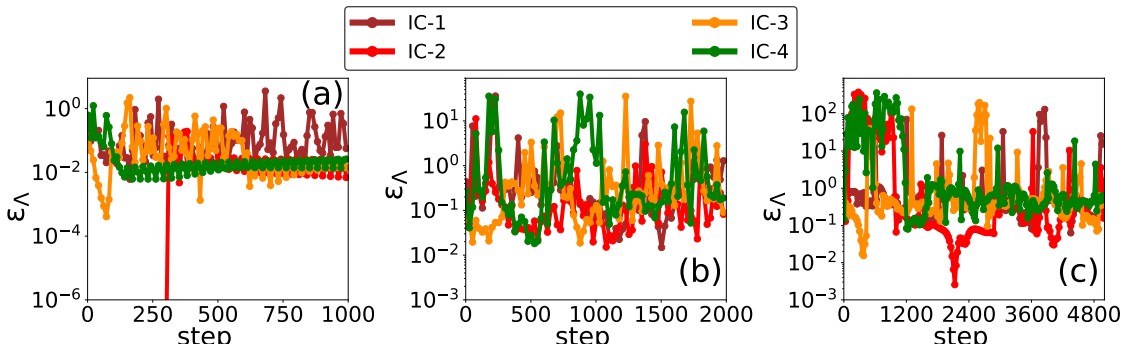

Figure 3: The evolution of $\epsilon_\Lambda$ at $J_y = 1.2$ using different initial conditions (IC): a random number state ("IC-1", brown line), an empty state with all spins pointing 'down' ("IC-2", red line), a full state with all spins pointing 'up' ("IC-3", orange line), and a state where each spin has components $\langle \sigma^x \rangle = 1$, $\langle \sigma^y \rangle = 1$, $\langle \sigma^z \rangle = -1$ ("IC-4", green line), for timesteps (a) $\delta t = 10^{-1}$, (b) $\delta t = 10^{-2}$, (c) $\delta t = 10^{-3}$. All other parameters are the same as in Fig. 1. Energies given in units of $\kappa = 1$.

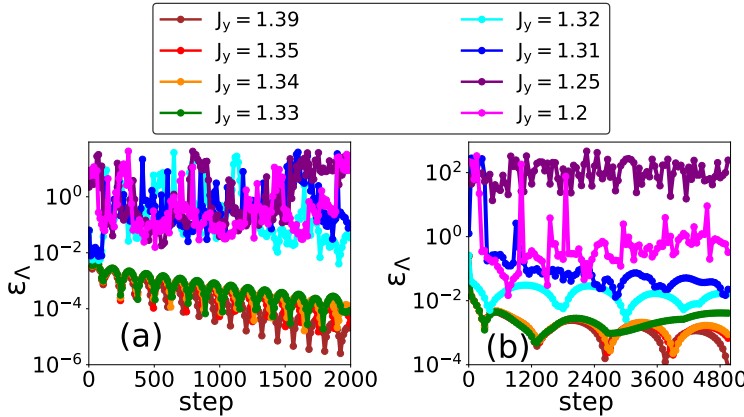

Figure 4: The evolution of $\epsilon_\Lambda$ at selected values of $J_y$ during the adiabatic parameter sweep from $J_y = 1.4$ (where a steady state exists) to $J_y = 1.2$ in steps of $\Delta J_y = 0.01$ and using timesteps (a) $\delta t = 10^{-2}$, (b) $\delta t = 10^{-3}$. All other parameters are the same as in Fig. 1. Energies given in units of $\kappa = 1$.

reach a steady state, then change $J_y$ in small steps, using the steady state at each value of $J_y$ as the initial state for the next value. This strategy can bypass highly entangled intermediate states where a very high $D$ may be needed. In our case, we start at $J_y = 1.4$, and gradually reduce $J_y$ in steps of $\Delta J_y = 0.01$ to $J_y = 1.2$. Figures 4(a,b) show the evolution of $\epsilon_\Lambda$ at selected values of $J_y$ during the parameter sweep, with timesteps $\delta t = 10^{-2}, 10^{-3}$ respectively. We observe that for $J_y \geq 1.33$, $\epsilon_\Lambda$ shows a decreasing trend, indicating that iPEPO finds a steady state, while for $J_y \leq 1.32$ we again find noisy oscillations. Smaller timesteps $\delta t$ and smaller sweeping steps $\Delta J_y$ (not shown) lead to the same conclusion.

A similar strategy is to start from a strong dissipation regime (i.e. large $\kappa$) where we know the steady state is approximately factorizable, and then perform an adiabatic sweep to lower $\kappa$. Figures 5(a,b) show the time evolution of $\epsilon_\Lambda$ at $J_y = 1.2$ for selected values of $\kappa$, in a sweep starting from $\kappa = 8$ reducing $\kappa$ in steps of $\Delta \kappa = 0.1$, with timesteps $\delta t = 10^{-2}, 10^{-3}$ respectively. Similarly to the $J_y$ sweep, we find a steady state exists for $\kappa \geq 5.2$, but beyond this we again find noisy oscillations. To summarise this section, for those points where a steady state is not found, this result appears to be robust to a variety of initial states and simulation

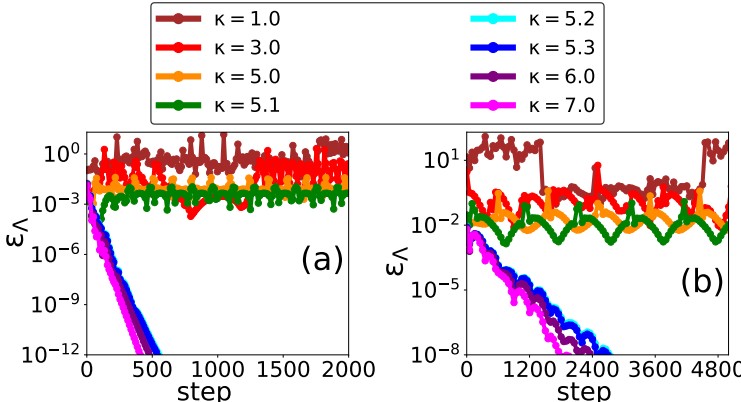

Figure 5: The evolution of $\epsilon_\Lambda$ at $J_y = 1.2$ for selected values of $\kappa$ during the adiabatic parameter sweep from a strong dissipation regime with $\kappa = 8$ to a weak dissipation regime with $\kappa = 1$ in steps of $\Delta\kappa = 0.1$ and using timesteps (a) $\delta t = 10^{-2}$, (b) $\delta t = 10^{-3}$. All other parameters are the same as in Fig. 1. Energies given in units of $\kappa = 1$.

protocols.

## 2.2 Effects of bond dimension

Figure 6 presents the effect of changing iPEPO bond dimensions D. Panels (a-c) show the time evolution of $\epsilon_\Lambda$ at $J_y = 1.2$ for timesteps $\delta t = 10^{-1}, 10^{-2}, 10^{-3}$ respectively. Each panel shows simulations performed using different bond dimensions $3 \leq D \leq 6$; no steady state is found for any of these values of $D$. Panels (d-f) show the same quantities but for $J_y = 1.5$, where a steady state is known to occur at $D = 4$. In this case, notably, while iPEPO reaches a steady state for $D = 3, 4$, no steady state is found for $D = 5, 6$. To explore this further, Panels (g-i) show the behavior at $J_y = 1.2$ for larger bond dimensions $10 \leq D \leq 15$. We observe that for $D = 12$ a steady state is found for timesteps $\delta t = 10^{-2}, 10^{-3}$. However, increasing the bond dimension further to $D = 14, 15$ leads again to noisy oscillations. These results suggest that while larger bond dimension may eventually yield a meaningful steady state, spurious steady states can arise at small bond dimension which then change as the bond dimension increases further. In addition, we note that while we can run the SU time evolution for $D = 15$, the CTM calculations for this bond dimensions would require over 128 GB of RAM, making it challenging to find observables without a support of high performance distributed-memory calculations and quantum symmetries. As noted above, the results shown in Fig. 6 do not depend on any CTM contraction, so this issue does not arise within the calculations we present.

## 3 Conclusion

From the results of the last section, we may conclude that the SU iPEPO algorithm at low bond dimensions is not always stable, reaching a steady state only in some parameter regimes, typically away from dissipative critical points. In other regimes, the algorithm failed to reach a steady state for all bond dimensions $D$ that we could access. Moreover, in some cases, even when a steady state is found for a given value of $D$, this may change as the bond dimension is increased, switching instead to noisy time-dependent dynamics. While we believe that there exists a value of $D$ allowing for a faithful representation of the steady-state density matrix

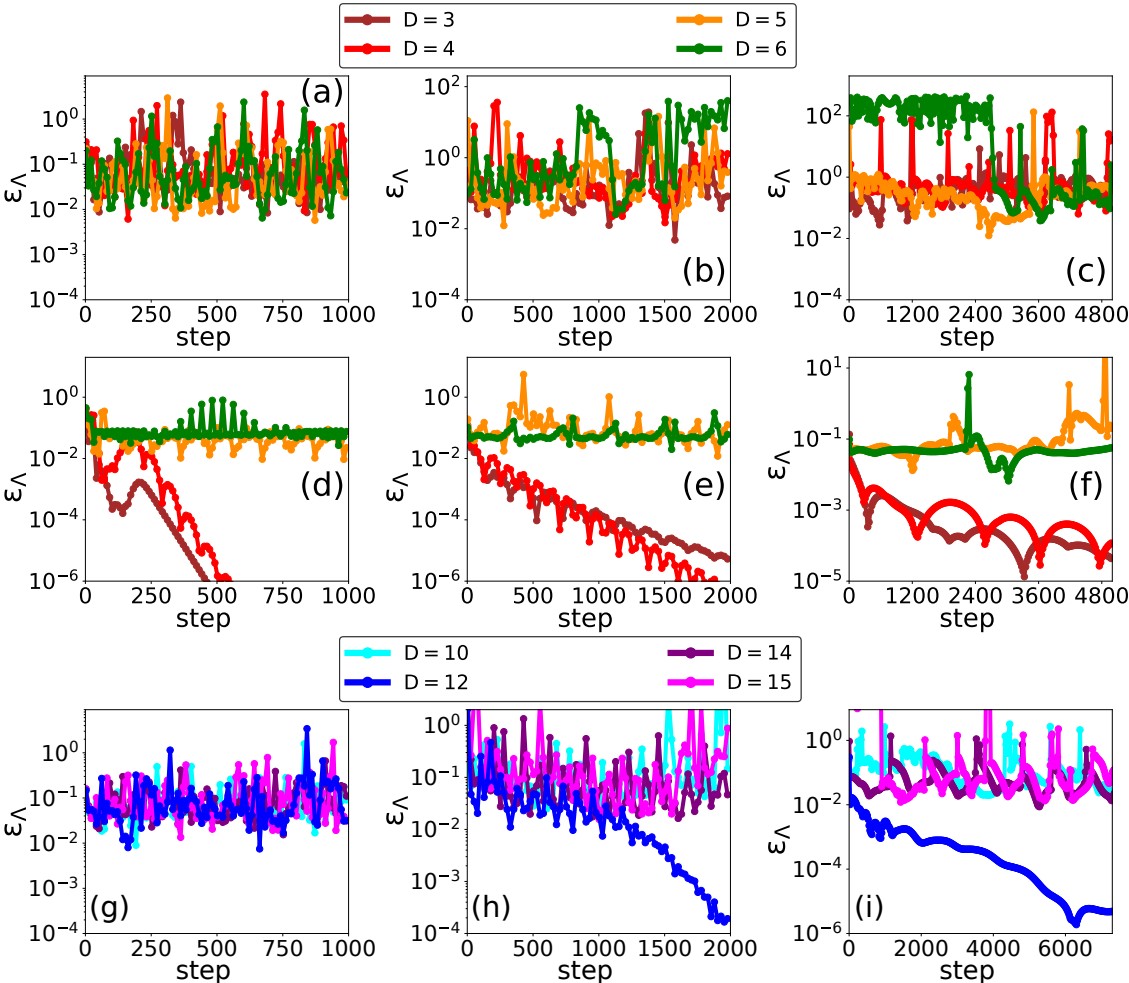

Figure 6: Evolution of $\epsilon_\Lambda$ with bond dimension at $J_y = 1.2$ (a-c,g-i) and $J_y = 1.5$ (d-f). Left column (panels a,d,g) are for timestep $\delta t = 10^{-1}$, middle (b,e,h) $\delta t = 10^{-2}$, and right (c,f,i) $\delta t = 10^{-3}$. Panels (a-f) show $D = 3, 4, 5, 6$, while panels (g-i) show $D = 10, 12, 14, 15$. All other parameters are the same as in Fig. 1. Energies given in units of $\kappa = 1$.

(when spatial correlations decay exponentially), this study is unable to conclude what *typical* value of the bond dimension is needed for a prototypical driven-dissipative lattice model. Overall, the results shown here suggest that a significant caution is required when extending the SU iPEPS algorithm in 2D to Liouvillian evolution. Below we discuss a number of alternative approaches which may be employed instead.

One alternative approach is to adapt the Full Update (FU) iPEPS algorithm [3, 24, 26, 37] to the Lindblad time evolution of mixed states. For closed systems, the FU scheme [3, 24, 38] achieves an optimal truncation by using a variational update scheme that computes the full environment at every step. Since the Liouvillian evolution involves non-Hermitian operators, an issue here is to find a reliable non-Hermitian algorithm that could substitute the alternating least-squares scheme used in the two-site variational minimization in the standard FU algorithm. There are, however, recent works on time evolution in closed systems [39–42] for which FU presents problems with stability, meaning SU can be more accurate. However, very recent work by McKeever and Szymanska [43] has shown that a variation on full update— full environment truncation—can indeed improve the stability of iPEPO.

A closely related idea is to consider a global variational search algorithm that targets the

null eigenstate $|\rho\rangle$ of either Liouvillian $\mathcal{L}$ or a Hermitian positive semidefinite object $\mathcal{L}^\dagger \mathcal{L}$; such approaches were successfully used in one dimension [15, 16], and extension of these methods to iPEPO has been discussed in Ref. [44]. Solving the variational problem with $\mathcal{L}^\dagger \mathcal{L}$ is particularly appealing since it allows reusing the standard and robust Hermitian optimization algorithms. However, even when $\mathcal{L} = \sum_l \mathcal{L}_{l,l+1}$ contains only nearest-neighbour terms, the product $\mathcal{L}^\dagger \mathcal{L} = \sum_{l,r} \mathcal{L}^\dagger_{l,l+1}\mathcal{L}_{l+r,l+r+1}$ will introduce highly nonlocal terms. While this is manageable in 1D [15], for 2D the nonlocal couplings may easily lead to unfeasibly large bond dimensions. This may perhaps be adressed by truncating the range of these nonlocal terms as has been discussed in 1D [45]. In addition, variational optimization iPEPO approaches would require computationally expensive tensor contractions involving both the iPEPS representing $|\rho\rangle$ and the iPEPO representing either $\mathcal{L}$ or $\mathcal{L}^\dagger \mathcal{L}$.

A more promising approach, also discussed in Ref. [44] may be to extend novel variational iPEPS techniques for ground state calculations in 2D introduced in Refs. [46, 47], optimizing iPEPS tensors using tangent space methods or by solving a local generalized eigenvalue problem. Notably, both approaches avoid the need to construct a full PEPO for the Hamiltonian. Adapting these algorithms to either $\mathcal{L}$ or $\mathcal{L}^\dagger \mathcal{L}$ could dramatically reduce the computational costs that limit the practical use of variational iPEPS methods. The global variational optimization could also offer a potentially much more robust way of finding the NESS of $\mathcal{L}$ than one could hope to achieve with the standard iPEPS algorithm relying on two-body updates.

# Acknowledgements

We acknowledge helpful discussions with Marzena Szymańska, Conor McKeever, and Andrew Daley. We are grateful for comments from Augustine Kshetrimayum on an earlier version of this manuscript.

**Author contributions**   The calculations presented here were performed by DK, using code developed by DK with contributions from AB. The project was initially conceived by JK and RF. All authors contributed to the writing of the manuscript.

**Funding information**   DK acknowledges support from the EPSRC Condensed Matter Centre for Doctoral Training (EP/L015110/1). JK, RF and AB acknowledge the Kavli Institute for Theoretical Physics, University of California, Santa Barbara (USA) for the hospitality and support during the early stages of this work; this research was supported in part by the National Science Foundation under Grant No. NSF PHY11-25915. AB acknowledges funding by LabEx PALM (ANR-10-LABX-0039-PALM).

# A   Implementation of iPEPO algorithm

As noted above, iPEPO is a simple extension of the iPEPS algorithm to nonequilibrium steady states of Lindblad superoperators. As with other extensions of tensor network methods [10] the main idea is to frame the density matrix evolution through superoperators applied to vectorized many body density matrices, which we denote $|\rho\rangle_\sharp$. In this appendix we first briefly summarize the iPEPS algorithm and then discuss its extension to open quantum systems. While these algorithms are standard and have been described in full by Orús in [3], we summarise them here to provide a self-contained description of the method that we apply to the open quantum systems.

## A.1 Summary of the iPEPS algorithm

The basic idea behind PEPS is to parameterize the quantum state tensor $\Psi_{k_1,k_2,\dots,k_N}$ by a two-dimensional array of interconnected rank-5 tensors (see Fig. 7). Each individual tensor represents a single site of the quantum many-body system, with one vertical leg corresponding to the local Hilbert space of dimension $d$, and four in-plane legs corresponding to the bonds between different lattice sites. We denote the bond dimension of PEPS by $D$, which limits the amount of entanglement that can be captured by PEPS.

For translationally invariant systems, one may use the infinite PEPS (iPEPS) ansatz [24] working directly in the thermodynamic limit. We can construct an iPEPS by choosing a unit cell and representing its sites with tensors. We will consider problems with a square unit cell. Since we use a Trotterised time evolution that propagates pairs of sites, we will need only two on-site tensors $A$ and $B$ to define iPEPS. We next describe the two main ingredients of the iPEPS approach: the imaginary time propagation of iPEPS and the calculation of the environment needed to extract observables.

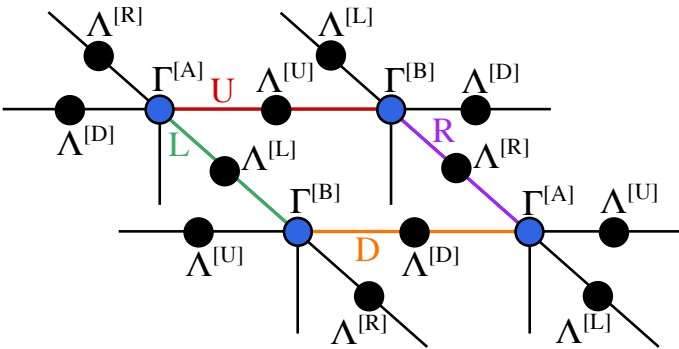

Figure 7: The iPEPS time evolution using Eq. (5) involves propagating four different bonds 'U', 'D', 'R', and 'L', indicated by different colours. The SU algorithm uses Vidal form with $\Gamma^{[A,B]}$ site tensors and $\Lambda^{[U,R,D,L]}$ diagonal bond matrices to represent iPEPS with a two-site unit cell.

### A.1.1 Time evolution: simple update

Time evolution can be performed by the Simple Update (SU) method, which follows essentially the same main steps as the imaginary time infinite Time Evolving Block Decimation (iTEBD) algorithm [48–51]. In two dimensions we perform Trotter decomposition by splitting our Hamiltonian into four terms $H_U$, $H_D$, $H_R$ and $H_L$, describing respectively the 'U' (up), 'D' (down), 'R' (right), and 'L' (left) bonds of the lattice:

$$H = H_U + H_D + H_R + H_L. \tag{4}$$

The first order Trotter decomposition of the time evolution operator $U(\delta\tau) = e^{-H\delta\tau}$ then reads:

$$U(\delta\tau) = e^{-\delta\tau H_U} e^{-\delta\tau H_R} e^{-\delta\tau H_D} e^{-\delta\tau H_L} + O(\delta\tau^2), \tag{5}$$

where $\delta\tau$ is the imaginary timestep. Similarly to iTEBD in one dimension, in SU we represent iPEPS using Vidal form: i.e. the iPEPS with a two-site unit cell is fully specified by two $\Gamma^{[A,B]}$ site tensors and four $\Lambda^{[U,R,D,L]}$ diagonal matrices that store the singular values of iPEPS bonds, as seen in Fig. 7. We denote the local Hilbert space dimension by $d$, and the bond dimension by $D$. The SU then consists of the following steps:

1. Absorb $\Lambda^{[R,D,L]}$ tensors on the external bonds into $\Gamma^{[A,B]}$ to obtain $Q^{[A,B]}$.

2. Decompose each of $Q^{[A,B]}$ into subtensors $v_{A,B}$ and $X_A$, $Y_B$ using an exact SVD or QR/LQ decompositions. The original rank-5 tensors $\Gamma^{[A,B]}$ had the dimensionality of $dD^4$ giving rise to a large computational cost $O(d^3D^9)$ of the update procedure. However, an update performed using the new rank-3 subtensors has a substantially reduced cost of $O(d^6D^3)$ since the dimensions of $v_{A,B}$ are considerably smaller and equal to $d \times dD \times D$ and $d \times D \times dD$ respectively.

3. Contract the two-body propagator $e^{-\delta\tau H_U}$ with $v_{A,B}$ and $\Lambda^{[U]}$ to form $\theta$ tensor.

4. Decompose $\theta$ tensor into $\tilde{v}_{A,B}$ and $\tilde{\Lambda}^{[U]}$ tensors using SVD. To prevent the bond dimension of our tensors from growing indefinitely, we must truncate $\tilde{v}_{A,B}$ and $\tilde{\Lambda}^{[U]}$ by retaining $D$ largest singular values and discarding the rest.

5. Recover the updated rank-5 tensors $\tilde{Q}^{[A,B]}$ by contracting the rank-3 subtensors $\tilde{v}_{A,B}$ with $X_A$, $Y_B$ respectively.

6. To restore $\Lambda^{[R,D,L]}$ on the external bonds, we divide each of $Q^{[A,B]}$ by $\Lambda^{[R]}$, $\Lambda^{[D]}$, and $\Lambda^{[L]}$. This procedure brings iPEPS back to its original Vidal form, with updated tensors $\tilde{\Gamma}^{[A,B]}$ and $\tilde{\Lambda}^{[U]}$ for each 'U' bond on the lattice.

All other steps are the same as in the one dimensional iTEBD algorithm. To find the ground state we propagate iPEPS for $N$ imaginary timesteps $\delta\tau$ until a steady state is achieved with respect to the spectrum of singular values in $\Lambda^{[U,R,D,L]}$.

The SU algorithm is both simple and very efficient, with computational cost $O(D^3 d^6)$ of the time evolution. However, SU is suboptimal since it employs local truncations without taking into account the full environment of a unit cell. For MPS, this issue can be resolved relatively easily by transforming the tensor network into a canonical form, which orthonormalizes other bonds surrounding the bond being truncated. This solution is not possible in 2D, since there is no known canonical form for PEPS. To achieve an optimal truncation, one must use a variational update scheme that computes the full environment at every step. This procedure, known as the Full Update (FU) [3,24], is considerably more expensive and bears the computational cost of $O(N\chi^3 D^6 + N\chi^2 D^8)$ where $N$ is the number of steps of the imaginary time evolution. In practice, SU has been applied extensively to various models and yields sufficiently accurate results for systems with large gaps and sufficiently short correlation lengths [52]. Due to its simplicity and efficiency, it allows shorter computation times and significantly higher bond dimensions than FU, and thus remains popular. However, the suboptimal truncation becomes an issue near quantum critical points when correlation lengths become long, and in these cases FU should be used instead.

### A.1.2 Contraction: corner transfer matrix

For the iPEPS representation to be of practical use, we must be able to extract expectation values from it. Unlike MPS where we could evaluate overlaps exactly at a polynomial cost, the exact contraction of two PEPS is an exponentially hard problem that scales as $O(e^L)$ with PEPS size $L$ [3]. Fortunately, there exist various computational algorithms that can perform this contraction approximately with high precision. For infinite systems, these methods typically proceed by computing the approximate environment of an iPEPS unit cell. This effective environment consists of a small set of tensors that represent the infinite tensor network surrounding the unit cell. Possibly the most successful technique for computing iPEPS environments is the corner transfer matrix (CTM) method [25,53,54], which will be the method we use. In this section we will explain the details of the CTM algorithm.

Since observables for quantum states involve the overlap of two copies of the state, the starting point for CTM is the contraction of two iPEPS, which produces an infinite 2D network made of reduced tensors $a$. Each reduced tensor $a$ results from the contraction of $\left[M^A\right]^\dagger$ and $M^A$ iPEPS tensors by their physical indices, except at the sites where any operator is applied, leading to a different reduced tensor $a_O$. Supposing the bond indices of $M^A$ had dimension $D$, the reduced tensors $a$ now have bonds of dimension $D^2$.

For simplicity of exposition, we will start by considering a one-site unit cell. However, methods based on Trotter decomposition into even and odd bonds modify the translational invariance from one-site to two-site. Therefore, in practice we always use the two-site version of the CTM algorithm. Let us now subdivide this network into a $1 \times 1$ unit cell made of $a$ tensors, and its environment that contains the remaining infinite tensor network in which the unit cell in embedded (see Fig. 8). The key idea is to represent the environment by a set of four corner matrices $\{C_{1,2,3,4}\}$, and four transfer tensors $\{T_{1,2,3,4}\}$ – these tensors are connected by new virtual bond indices of size $\chi$. Similarly to the bond dimension of MPS and PEPS, the environmental bond dimension $\chi$ is the parameter that controls the accuracy of the CTM approximation of environment. The goal of CTM algorithm is to obtain the environmental tensors by performing a series of coarse graining moves:

1. Initialize the CTM tensors, e.g. using a random-number initialization, or a mean field environment with $\chi = 1$.

2. Perform four coarse graining moves in the left, right, up, and down directions. The left move involves the following steps, illustrated graphically in Fig. 8(a):

3. Insertion: insert an extra column into the CTM network that contains the unit cell tensor $a$, and the transfer tensors $T_{1,3}$.

4. Absorption: absorb the new column into the left side of the CTM environment by contracting their respective tensors. This increases the environmental bond dimension by $\chi \to D^2\chi$: to prevent the bonds from growing indefinitely we must implement an appropriate truncation scheme.

5. Renormalization: truncate the environmental tensors by inserting appropriate isometries $ZZ^+ = I$ that reduce the bond dimensions $D^2\chi \to \chi$ by projecting onto a relevant subspace, as shown in Fig. 8(b).

6. Repeat steps (2-5) to let the CTM environment grow in all four directions until it converges. Convergence is typically achieved when the eigenspectrum of each corner matrix $\{C_{1,2,3,4}\}$ reaches the fixed point.

The CTM algorithm for a two-site unit cell, containing two $a$ and $b$ tensors, follows the same steps as the one-site algorithm outlined above. The environment is now specified by a set of four corner matrices $\{C_{1,2,3,4}\}$, and eight transfer tensors $\{T^a_{1,2,3,4}, T^b_{1,2,3,4}\}$. The main difference is that in the 'Insertion' step we now insert two new columns instead of just one, as shown in Fig. 8(c). There are now two 'Absorption' and 'Renormalization' steps in the algorithm: the absorption of each column is followed by renormalization to reduce the bond dimension $D^2\chi \to \chi$. The two-site algorithm also needs two types of isometries $ZZ^+ = I$ and $WW^+ = I$ in the 'Renormalize' step, to obtain renormalized transfer tensors $\tilde{T}^{a,b}_4$ and corner tensors $\tilde{C}_{1,4}$ in Fig. 8(d).

Clearly, the crucial step of CTM algorithm is calculating the isometries. Several different methods exist, for instance the ones described in Refs. [25, 37, 55], which we have implemented and tested in the process of developing our iPEPS code. The prescription we have

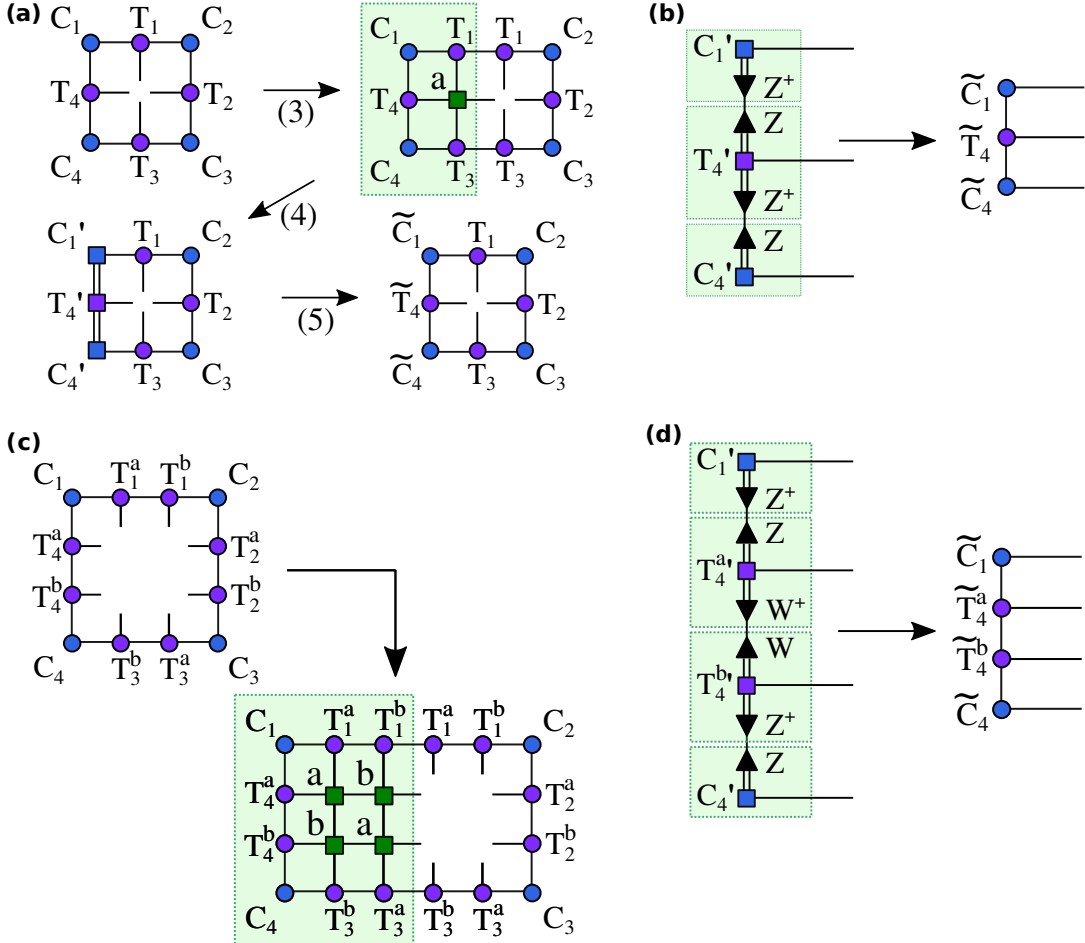

Figure 8: (a) The main steps (3-5) of the left move of the CTM algorithm for a one-site unit cell containing tensor $a$. (b) The renormalization step (5) is done by inserting isometries $ZZ^+ = I$ into the left edge of CTM network. (c) The insertion step of the left move of the CTM algorithm for a two-site unit cell containing tensors $a$, $b$. (d) The renormalization step for a two-site unit cell is done by inserting two types of isometries, $ZZ^+ = I$ and $WW^+ = I$.

found to work best was the one in Ref. [37], which achieved a smoother convergence and a more efficient representation of the environment than its predecessors. Figure 9 illustrates graphically the calculation of $Z, W$ isometries to be used in the 'Renormalize' step of the left move. In Fig. 9(a,b) we compute $W, W^+$ isometries to be inserted into the bonds split by the 'cut-1'. In the first stage in Fig. 9(a), we contract the lower and upper parts of the network, producing the tensors $Q_A$ and $Q_B$ respectively. In the second stage, also in Fig. 9(a), we decompose $Q_A$ and $Q_B$ using an exact SVD to obtain the $R_{A,B}$ and $V^\dagger_{A,B}$ tensors. In the third stage in Fig. 9(b), we form the product $I = R_A \left[ R_A^{-1} R_B^{-1} \right] R_B$, and decompose $\left[ R_A^{-1} R_B^{-1} \right] = U \Lambda V^\dagger$ using SVD, this time truncating to the $\chi$ dominant singular values. The $R_{A,B}$ matrices and the SVD matrices are then combined in the symmetrized fashion $I \approx \left[ R_A U \Lambda^{1/2} \right] \left[ \Lambda^{1/2} V^\dagger R_B \right] = WW^+$ to construct the isometries $W = \left[ R_A U \Lambda^{1/2} \right]$ and $W^+ = \left[ \Lambda^{1/2} V^\dagger R_B \right]$. To obtain the $Z, Z^+$ isometries for the bonds split by the 'cut-2', we perform a translationally-invariant shift by inserting two extra rows of tensors in the green box, as shown in Fig. 9(c). Contracting the lower and upper parts of the network then gives tensors $Q_{A,B}$ for the cut-2. We can now compute $Z, Z^+$ repeating exactly the same steps as for the $W, W^+$ before. Once all isometries $Z, Z^+$ and $W, W^+$

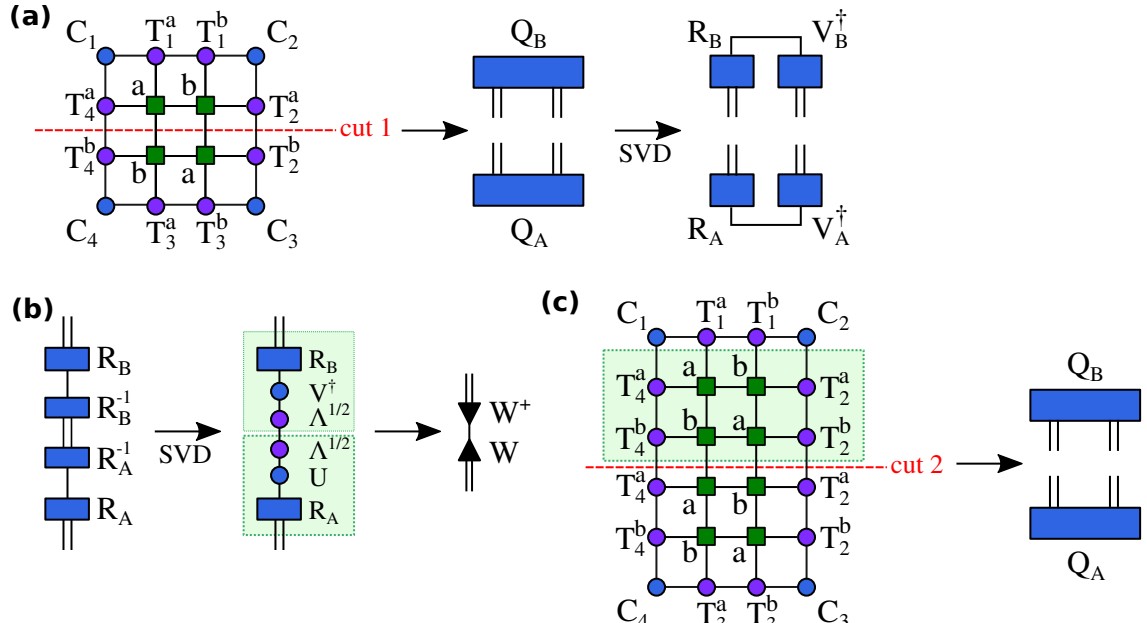

Figure 9: (a,b) The steps to compute $W, W^+$ isometries for the bonds split by the 'cut-1'. (c) To compute $Z, Z^+$ isometries for the bonds split by the 'cut-2', we perform a translationally-invariant shift by inserting two extra rows of tensors in the green box, and repeat the steps in (a).

are available, one can finally use them to carry out the renormalization in Fig. 8(d).

The CTM algorithm described above has a computational complexity of $O(\chi^3 D^6 + \chi^2 D^8)$. Once we have found a converged CTM environment, it can be used to compute various observables.

## A.2 Extension to iPEPO

To extend the above method to an open quantum system, as noted above, one may first represent the density matrix $\rho$ as a Projected Entangled Pair Operator (PEPO), and then reshape (vectorize) it into a PEPS $|\rho\rangle_\sharp$ by combining both physical indices at each site. The problem of computing NESS for an infinite 2D lattice thus becomes equivalent to the problem of finding the ground states of Hamiltonians using the SU iPEPS algorithm, by replacing imaginary time Hamiltonian propagation with the real time Liouvillian propagation. To distinguish between the iPEPS representing wavefunctions and vectorized density operators, we will refer to the density matrix version as the iPEPO algorithm.

The main steps of the iPEPO algorithm are the same as in Sec. A.1, except for two differences that we discuss next. The first difference is the propagator. The imaginary time two-body propagators $U_\alpha(\delta\tau) = e^{-\delta\tau H_\alpha}$ with Hamiltonian $H_\alpha$ for a bond $\alpha \in \{U, R, D, L\}$ are replaced by the real time two-body propagators $U_\alpha(\delta t) = e^{-\delta t \mathcal{L}_\alpha}$, where $\mathcal{L}_\alpha$ is the two-body Liouvillian for a bond $\alpha \in \{U, R, D, L\}$ and $t$ is the real time. The second difference is that observables are calculated using $\langle O \rangle = \text{Tr}[O\rho]$, instead of $\langle O \rangle = \langle \Psi|O|\Psi \rangle$. Similarly, the correct normalization $\text{Tr}[\rho] = 1$ in contrast to $\langle \Psi|\Psi \rangle = 1$. As such, when extracting observables from iPEPO, the reduced tensors that are contracted to find the environment come from tracing out local indices instead of computing inner products. We may then apply the CTM method from Sec. A.1.2 to observables.

We have implemented our iPEPO code in Fortran [36], including the CTM algorithm, the functionality required for computing local observables and two-point correlators, the SU proce-

dure for both NESS and ground state calculations, as well as the FU procedure for ground state calculations (for benchmarking purposes only). In our calculations we gradually decrease the time step $\delta t$ during the simulation to reduce the effects of Trotter error while keeping the computational cost low. To determine when the calculation has reached a steady state we require that the spectrum of singular values contained in each diagonal bond matrix $\Lambda \in \left\{ \Lambda^{[U,D,R,L]} \right\}$ in Fig. 7 stops changing within some accuracy $\epsilon$. More specifically, we take the largest difference between singular values in diagonal matrices $\Lambda_n$ and $\Lambda_{n-1}$, at timesteps $n$ and $n-1$ respectively, rescaled by the largest singular value $|\Lambda_n|_{\max}$ and by timestep size $\delta t$:

$$\epsilon_\Lambda = \frac{|\Lambda_n - \Lambda_{n-1}|_{\max}}{\delta t \, |\Lambda_n|_{\max}}. \tag{3}$$

For a steady state we expect $\epsilon_\Lambda$ to approach zero (or more precisely, a value depending on machine precision), as the eigenvalue spectrum should cease changing. To determine numerically when to stop, we define a steady state as being reached when $\epsilon_\Lambda < \epsilon$ for each $\Lambda \in \left\{ \Lambda^{[U,D,R,L]} \right\}$.

### A.3   Benchmarking with ground state calculations

Due to the similarity between the iPEPS and iPEPO algorithms, we may benchmark our implementation against the known numerical results from ground state calculations of two models: the transverse field Ising model, and the hardcore Bose-Hubbard model.

**Transverse-field Ising model**   Our first test problem is the transverse field Ising model on an infinite square lattice,

$$H = -J \sum_{\langle i,j \rangle} \sigma_i^z \sigma_j^z - g \sum_i \sigma_i^x. \tag{6}$$

Here $\sigma_i^{x,z}$ are Pauli matrices at site $i$, $J$ is the nearest-neighbour coupling between spins, and $g$ is the transverse magnetic field along the $x$ axis. The ground state of this model exhibits a second order phase transition between a paramagnetic phase at large $g$, and a ferromagnetic phase at small $g$; the order parameter of this transition is the longitudinal magnetization $M_z = \langle \Psi_{GS} | \sigma^z | \Psi_{GS} \rangle$. We show results in units where $J = 1$.

Figure 10(a,b) shows the longitudinal magnetization $M_z$ and transverse magnetization $M_x$ as a function of transverse field $g$ in the vicinity of phase transition, for different values of iPEPS bond dimension $D$. Our implementation of iPEPS reproduces accurately both the SU and FU results reported in Refs. [24–26]. As expected, the SU and FU calculations match well far from the critical point where correlations are short-ranged, and the results converge fast with increasing values of $D$. Near the critical point FU becomes considerably more accurate than SU at a given bond dimension. That is, due to the diverging correlation length, a much higher value of $D$ is needed for SU to achieve the same level of accuracy as FU with $D = 2, 3$. As seen from Fig. 10(a), the FU calculation with $D = 3$ predicts the critical point around $g = 3.05$, in good agreement with previous iPEPS results in Refs. [24–26] and Quantum Monte Carlo results in Ref. [56].

**Hardcore Bose-Hubbard model**   Our second test case is the Bose-Hubbard model (BHM) on an infinite square lattice,

$$H = -J \sum_{\langle i,j \rangle} \left( a_i^\dagger a_j + \text{H.c.} \right) - \mu \sum_i a_i^\dagger a_i, \tag{7}$$

where the occupations are restricted to 0 or 1 bosons on each lattice site. Here, $a_i^\dagger$, $a_i$ are hardcore bosonic creation and annihilation operators at site $i$, satisfying the commutation relation

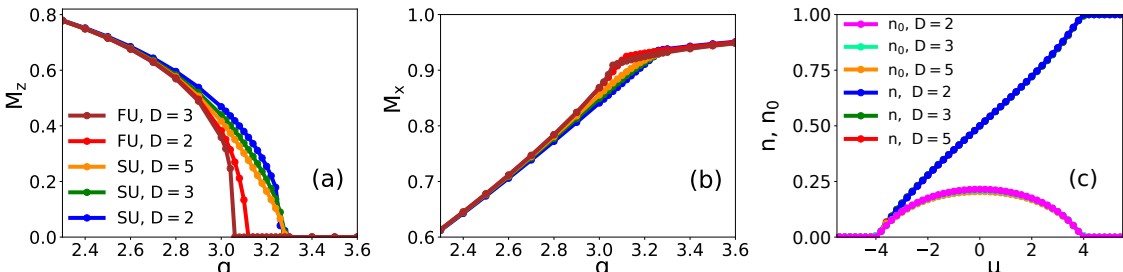

Figure 10: Ground state results benchmarking our iPEPS implementation. Panels (a,b) show (a) the longitudinal magnetization $M_z$ and (b) transverse magnetization $M_x$ as a function of transverse field $g$ (in units of $J$) in the vicinity of phase transition of the transverse field Ising model. These are computed for different iPEPS bond dimensions $D$ using both SU and FU as indicated. Panel (c) shows number density of bosons $n$ per lattice site and the condensate fraction $n_0$ for the hardcore Bose-Hubbard model as a function of chemical potential $\mu$ (in units of $J$), computed for different iPEPS bond dimensions $D$ using SU.

$\left[a_i^\dagger, a_j\right] = (1 - 2a_i^\dagger a_i)\delta_{ij}$. $J$ is the hopping rate between adjacent sites, and $\mu$ is the onsite chemical potential. This model undergoes a second order phase transition between the superfluid and the Mott insulator phases at the critical value of $\mu/J$. As before, we present results in units where $J = 1$. Figure 10(c) shows the number density of bosons $n = \langle\Psi_{GS}|a^\dagger a|\Psi_{GS}\rangle$ and the condensate fraction $n_0 = |\langle\Psi_{GS}|a|\Psi_{GS}\rangle|^2$ as a function of chemical potential $\mu$, for different bond dimensions $D$ of SU iPEPS. Again, our iPEPS calculations reproduce accurately the ground state results of Refs. [26, 57, 58].

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
