# Peer review of "On the stability of the infinite Projected Entangled Pair Operator ansatz for driven-dissipative 2D lattices"

_SciPost Physics Core, doi:SciPost Phys. Core 4, 005 (2021)_

## Round 1 · Referee Report · Anonymous · 2021-1-21

Report

The manuscript ‘On the stability of the infinite Projected Entangled Pair Operator ansatz for driven-dissipative 2D lattices’ by D. Kilda et al reproduces the earlier work of Kshetrimayum et al (Nat. Comm. 8, 1291 (2017)) and discusses the stability of the iPEPO algorithm at certain parameter regimes by looking at the dissipative XYZ model. They look at this model for different bond dimensions of the iPEPO and different initial states.

The study of open quantum system in 2D is a very challenging and important problem in physics and such works suggest that much remains to be done towards the numerical investigation of dissipative systems specially in the critical regimes. The paper is short but clear. While the findings of the paper are not surprising, I recommend its publication in SciPost Physics. However, I have some comments for which I would like to seek clarification from the authors.

1) Looking at Figure 1, the red regions correspond to the parameter regime where the iPEPO does not reach a steady state. They also correspond to the critical regimes which are difficult for Tensor networks in general even in closed systems. Similar result has been found in the original work of Nat. Comm. 8, 1291 (2017) where the authors compute the expectation value of the vectorized Liouvillian operator. Can the authors comment more elaborately on this?

2) In Fig. 1 again, the authors present results obtained using different Trotter steps. Can the authors explain why the results are so different for these particular regions? The trotter error coming from large step is of the order of \delta t^2 and this can be minimized by taking higher order trotterization. I recommend the authors to implement this and see if taking larger trotter steps help the calculation and if so, why.

3) In Fig. 2, the singular values for different parameter regimes are shown. However, it is not clear to me why the authors have shown only one singular value for each parameter. Is this the largest singular value? What happened to the rest of the singular values?

4) In Fig. 6, the authors take different bond dimensions and comment that some of the simulations could get unstable with increasing bond dimension. For simulations with D=10, 12, 14, 15, what is the bond dimension of the environment used here? Are these results correctly converged with the bond dimension of the environment?

In general, I feel that the authors present the result but does not offer much explanation or interpretation of the results. For example, why do different trotter step give different results, how their figure of merit is related to the results obtained by Kshetrimayum et al, how will the full update provide better stability, etc. I suggest the authors to take care of these things before publishing the manuscript.

---

## Round 2 · Author Response

Dear Editor

We thank the referee for their comments on our manuscript, which have helped us explain a number of points which were previously unclear. The referee recommends publication of our manuscript after the issues they raised are addressed. Below, we respond to these points, and describe the changes made to our manuscript.

One aspect that we find important to address is the score on the originality that was assigned in the report. We do agree with the referee on this point, in that our work addresses a method and model that have been considered previously. However, despite this having been addressed previously, we believe our work is useful for the community, hence our reason for submitting it. The main motivation of the present work is to understand the limits of the interesting method developed by Kshetrimayum et al. This, in our opinion, is a necessary step to improve it and it will be, in our opinion, useful material for the groups working on this area.

We hope that with the changes described below, our manuscript may be suitable for publication.

Yours,
The Authors

---

## Round 2 · List of Changes

Response to the referee:

> The manuscript ‘On the stability of the infinite Projected Entangled Pair
> Operator ansatz for driven-dissipative 2D lattices’ by D. Kilda et al
> reproduces the earlier work of Kshetrimayum et al (Nat. Comm. 8, 1291
> (2017)) and discusses the stability of the iPEPO algorithm at certain
> parameter regimes by looking at the dissipative XYZ model. They look at
> this model for different bond dimensions of the iPEPO and different initial
> states.

> The study of open quantum system in 2D is a very challenging and important
> problem in physics and such works suggest that much remains to be done
> towards the numerical investigation of dissipative systems specially in the
> critical regimes. The paper is short but clear. While the findings of the
> paper are not surprising, I recommend its publication in SciPost
> Physics. However, I have some comments for which I would like to seek
> clarification from the authors.

> 1) Looking at Figure 1, the red regions correspond to the parameter regime
> where the iPEPO does not reach a steady state. They also correspond to the
> critical regimes which are difficult for Tensor networks in general even in
> closed systems. Similar result has been found in the original work of
> Nat. Comm. 8, 1291 (2017) where the authors compute the expectation value
> of the vectorized Liouvillian operator. Can the authors comment more
> elaborately on this?

We agree that our manuscript did not previously make this point sufficiently clearly. Indeed Kshetrimayum et al. point out possible problems in reaching the steady state, and the aim of our work is to explore the origin of these problems. Specifically, the aim of our figure 1 is to note that in these regions, the dynamics does not ever reach a steady state. This is the starting point of our analysis that will be developed in the subsequent sections. We have modified the text to clarify that possible issues related to stability were mentioned in the work of Kshetrimayum et al.; see sentences starting "The red regions in our figure..." on page 3.

> 2) In Fig. 1 again, the authors present results obtained using different
> Trotter steps. Can the authors explain why the results are so different for
> these particular regions?

The difference between 1(a) and 1(b) is not merely the Trotter step. In 1(a), in the white regions, the time evolution is performed until a steady state is reached. i.e., the time evolution continues until the state stops changing. In 1(b), as noted in the figure caption, the time evolution is run for a fixed number of steps and then stops. As such, there is no guarantee that the state reached is a steady state. This is the key difference between the two figures.

While this point was noted in the figure caption, we have clarified the discussion of this in the text, see sentence starting "However, because the simulation...", at the end of page 3.

> The trotter error coming from large step is of
> the order of \delta t^2 and this can be minimized by taking higher order
> trotterization. I recommend the authors to implement this and see if taking
> larger trotter steps help the calculation and if so, why.

As noted above, Fig. 1(b) does not indicate that a larger Trotter step improves stability, it instead shows that stopping the simulation artificially early produces a smooth but erroneous curve. Regarding investigation of the effect of timestep, we note that this was explored in Fig. 2, where the three panels show the behaviour for three different timesteps. From the results presented in that figure, there is no evidence that larger timestep changes the stability, and for that reason, we do not expect higher order Trotterisation to improve the behaviour of the current algorithm.

> 3) In Fig. 2, the singular values for different parameter regimes are
> shown. However, it is not clear to me why the authors have shown only one
> singular value for each parameter. Is this the largest singular value? What
> happened to the rest of the singular values?

The measure epsilon_Lambda used in Fig. 2 and throughout is the largest change of singular value. Since the change of all other singular values is by definition smaller, then if epsilon_Lambda vanishes, this indicates that all singular values have stopped changing. We thus choose this parameter as it is both a necessary and sufficient condition for a steady state that this measure should vanish. We realise that the definition of epsilon_Lambda may have been unclear as it was previously only given in an appendix. To make this clearer, we have moved the definition of epsilon_Lambda to page 4, just before the start of section 2.1, and have referred to this definition in the caption of Figure 2.

> 4) In Fig. 6, the authors take different bond dimensions and comment that
> some of the simulations could get unstable with increasing bond
> dimension. For simulations with D=10, 12, 14, 15, what is the bond
> dimension of the environment used here? Are these results correctly
> converged with the bond dimension of the environment?

The bond dimension of the environment does not enter in to the calculation of the quantities plotted in Figures 2 - 6. This is because the quantity plotted in all these figures is purely dependent on the singular value spectrum of the iPEPO, which is purely controlled by the SU time evolution and does not require any contraction. i.e., no environment contraction is required to produce this figure, so there is no question of convergence with bond dimension of the environment.

This point was already mentioned on page 4 "As we discuss further below, the failure to reach a steady state that we observe occurs specifically in the SU time evolution. That is, it is completely unaffected by the CTM contractions needed to compute observables." We have further clarified that text to highlight that Figures 2 - 6 do not depend on the CTM contraction (and to add a missing definition of the acronym CTM). We have also added a reminder of this point on page eight.

> In general, I feel that the authors present the result but does not offer
> much explanation or interpretation of the results. For example, why do
> different trotter step give different results, how their figure of merit is
> related to the results obtained by Kshetrimayum et al, how will the full
> update provide better stability, etc. I suggest the authors to take care of
> these things before publishing the manuscript.

The points summarised here by the referee about Trotter timestep and the definition of the quantity epsilon_Lambda have been addressed above. Regarding the potential of alternate algorithms, we note that since the submission of our manuscript, a new preprint "Dynamics of two-dimensional open quantum lattice models with tensor networks" by Conor Mc Keever and Marzena H. Szymańska (arXiv:2012.12233) has been submitted, showing that an alternate update step does indeed significantly improve the stability of the algorithm. We have updated our discussion of this point to cite this new preprint which addresses far more thoroughly this idea.

You are currently on this page

Resubmission scipost_202012_00006v2 on 9 February 2021

---

## Editorial Decision

published